# Detection of opening motion characteristics in DC circuit breakers based on machine vision

**Zhaoyu Ku[1], Jinjin Li[2], Dongheng Li[1], Huajun Dong[1]***

**1** School of Mechanical Engineering, Dalian Jiaotong University, Dalian, China, **2** School of Automation and Electrical Engineering, Dalian Jiaotong University, Dalian, China

* xjzbzz@djtu.edu.cn

**Data Availability Statement:** All relevant data are within the article and its Supporting information files.

## Abstract

A circuit breaker is a crucial component in power systems, and its operation is essential for evaluating its interruption performance. However, electromagnetic interference often affects sensor accuracy. To address this issue, this paper investigates a non-contact measurement technique for assessing the motion characteristics of circuit breakers. A motion detection method based on Franklin moments is proposed. A synchronous image acquisition platform was established using high-speed cameras to capture the motion of 252kV circuit breakers. The captured images are preprocessed, with coarse edges extracted using the Laplacian algorithm. Franklin moment convolution calculations are then applied to determine sub-pixel coordinates of the image edges based on these coarse edges. By analyzing the frame-to-frame variations of these sub-pixel coordinates, the opening motion characteristics of the circuit breaker are extracted. This method can detect the vibration parameters and bouncing phenomenon of circuit breaker motion machine in millisecond level, and the accuracy is 0.01 mm. These findings offer valuable insights for future research on circuit breaker performance.

## Introduction

Due to its advantages such as minimal transmission loss and extensive transmission range, flexible DC transmission is extensively employed in the integration of new energy grids, urban power supply, and grid interconnection [1, 2]. The breaking capacity of the DC circuit breaker holds paramount significance in ensuring the reliable operation of the DC network, as it serves as a pivotal safety apparatus within the DC transmission system [3, 4]. The motion characteristics of the circuit breaker are directly reflected in its breaking condition, which is significant when the breaking performance of the circuit breaker is analyzed. As key components, the repulsion disc and nozzle allow the breaking ability of the circuit breaker to be analyzed from various angles, and the motion characteristics to be studied [5–7]. Further research into the motion characteristics of the circuit breaker's repulsion panel and nozzle is deemed essential for the safe operation of the power system.

The sensor method is considered the most effective approach for detecting the motion characteristics of circuit breakers. The moving structure of the circuit breaker is equipped with

**Funding:** This work was supported by the Natural Science Foundation of Liaoning Province under Grant 2024-BS-200 to J.L, and the Fundamental Research Funds for the Provincial Universities of Liaoning under Grant LJ212410150060 to J.L, and the Scientific Research Fund of Liaoning Provincial Education Department under Grant LJKMZ20220835 to H.D. The funders had no role in study design, data collection and analysis, decision to publish, or preparation of the manuscript.

**Competing interests:** The authors have declared that no competing interests exist.

various sensors, including acceleration, displacement, and grating sensors. However, it is inevitable that the integrity of the circuit breaker structure will be damaged by the sensor, which will have a seriously detrimental effect on its safe operation [8–10]. Furthermore, as the nozzle of the circuit breaker is situated in a closed high-pressure environment with strong electromagnetic interference, sensors cannot be installed within this internal environment. Therefore, there is an urgent need for a contactless detection method capable of accurately measuring the motion characteristics of the circuit breaker.

The advancement of image processing technology has enabled its use for detecting the motion characteristics of circuit breakers [11, 12]. The author in [13] and [14] employed a high-speed camera to collect and analyze images of key components of the circuit breaker to determine the opening and closing speeds. The authors in [15] proposed an algorithm for detecting circuit breaker motion using image block matching of moving images. A high-speed camera records the motion of an auxiliary mark on the tie rod or shaft, and the proposed algorithm analyzes its motion characteristics. The author in [16] combined image technology with finite element simulation. By integrating simulation results with measurements obtained from a high-speed camera, the static and dynamic characteristics of the short-circuit release force of the circuit breaker were analysed. The author in [17] proposed a non-contact testing method for the deformation characteristics of circuit breaker springs using image matching tracking technology. By predicting the target position of subsequent frames based on the positional correlation of preceding and following frames, the recognition area is dynamically adjusted according to these predictions. This approach helps eliminate interference from similar structures, enhancing both the matching speed and accuracy. Given the non-contact nature of circuit breakers, it is not surprising that considerable time and effort have been devoted to researching this area by scholars, resulting in extensive findings. Currently, the focus of research into non-contact motion detection methods for high-voltage circuit breakers is placed on indirect detection. By analyzing key structures of circuit breakers, the motion characteristics are inferred.

This paper offers a comprehensive and intuitive analysis of the breaking time and performance of high-voltage circuit breakers. It involves the collection and analysis of motion images captured during the circuit breaker's breaking process. A synchronous triggering loop is developed to align the image time axis with the circuit breaker's operation, enabling a precise examination of the circuit breaker's response time. The paper introduces a motion detection method based on Franklin moments, detailing the derivation of the sub-pixel edge detection principle and the calculation of a 5×5 template. This method is used to process and analyze the motion images of the circuit breaker's repulsion disc and nozzle. The study generates motion characteristic curves and vibration curves of the contact during breaking, offering robust technical support for investigating breaker failure, breaking time, and performance.

## Experimental and methods

The 252kV high-voltage circuit breaker is used as an example in this paper, and an image acquisition platform is set up. A method for detecting motion characteristics is then proposed, and the motion characteristics of the circuit breaker are analyzed in detail.

### Design of the experimental

To verify the effectiveness of the proposed algorithm in detecting motion in circuit breakers, this study uses a 252 kV high-speed open circuit breaker as a case study. Motion images of the repulsion disc and the contact external nozzle of the circuit breaker are captured. The actuator employed is a high-power hydraulic actuator known for its smooth operation. The circuit breaker's contact is encased by the nozzle and is rigidly connected. During the breaking

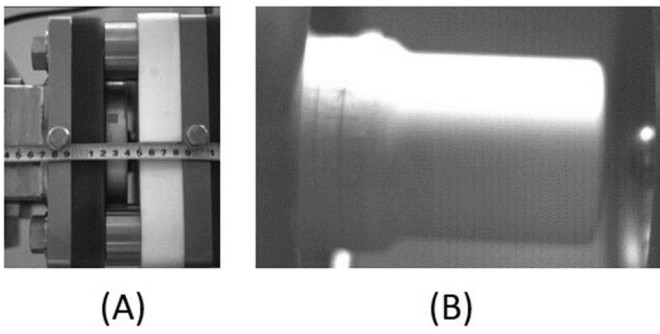

**Fig 1. Experimental target image.** (A) is the target of the repulsion disc of the operating mechanism. (B) is the image of the nozzle at the fracture.

process, the nozzle's jet extinguishes the arc, making direct observation of the arc conditions impossible. Given the structure of the circuit breaker, the contact and nozzle can be treated as a single rigid body with identical motion characteristics. Thus, by examining the motion characteristics of the nozzle, the contact's motion characteristics can be inferred. This study focuses on the nozzle's opening speed and vibration during the opening process, as well as on the repulsive force disc, with the specific style illustrated in Fig 1.

To acquire instantaneous image sequences of the circuit breaker's breaking movement, an image acquisition platform was built using a high-speed camera. This platform includes the camera, an optical lens, a complementary light device, a camera bracket, and a laptop computer. The high-speed camera has a maximum resolution of 1080×720 and a full frame rate of 12.68 kHz, resulting in an inter-frame time of approximately $7.886 \times 10^{-2}$ ms. The opening and closing movements of the high-voltage circuit breakers were controlled by the KOCOS system. To research the fault segmentation time of the circuit breaker, synchronous triggering of the high-speed camera and the circuit breaker is required. In this experiment, signals were simultaneously sent to the camera and the circuit breaker through a unified control system. The break signal was sent to the camera in real time to ensure synchronization between the high-speed camera and the circuit breaker. The circuit setup and experimental procedure are illustrated in Fig 2.

## Circuit breaker motion characteristic testing method

To detect the motion characteristics of high-voltage circuit breaker contacts, this paper presents a circuit breaker motion detection system based on Franklin moments. The system analyzes the changes in pixel coordinates of the target point of the circuit breaker structure between adjacent frames to determine the motion condition and obtain the mechanical motion characteristics of the circuit breaker.

**Definition of Franklin function and Franklin moments.** Franklin moments are a family of continuous orthogonal functions on $L^2[0,1]$, using the Franklin function. This function is derived from a linearly uncorrelated set of functions through a process of orthogonalization [18, 19]. The linearly uncorrelated set $\{a_n(x), 0 \leq x \leq 1\}$ is shown in Eq (1).

$$\begin{cases} a_i(x) = (x - a_i)_+, i = 2, 3, \cdots \\ a_0(x) = 1, a_1(x) = x \\ a_i = \dfrac{(2i - 1 - 2^j)}{2^j} \end{cases} \tag{1}$$

Where, $i$ is a positive integer, and $j$ is the highest value that satisfies $2j \leq 2i$-1.

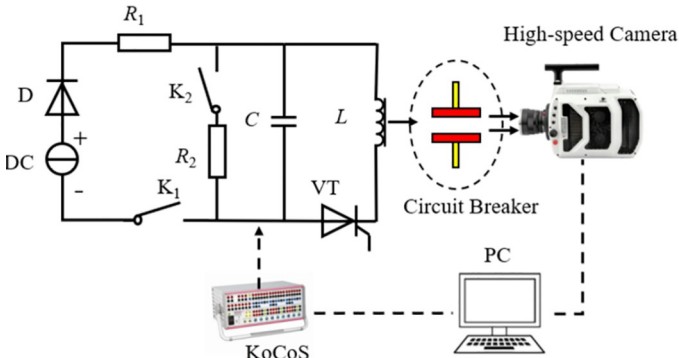

**Fig 2. Schematic diagram of the experimental platform.** DC and D constitute the experimental current source, $R_1$ is the charging resistance, $R_2$ is the discharge resistance, $L$ is the repulsive mechanism inductance, C is the drive loop capacitance, VT is the drive loop control thyristor, $K_1$ is the charging contactor, $K_2$ is the discharge contactor.

The linearly uncorrelated group is orthogonalized using the Gram-Schmidt process to obtain the Franklin function system. The first three expressions of the basis functions are as follows:

$$\varphi_0(x) = 1, 0 \leq x \leq 1 \tag{2}$$

$$\varphi_1(x) = \sqrt{3}(2x - 1), 0 \leq x \leq 1 \tag{3}$$

$$\varphi_2(x) = \begin{cases} \sqrt{3}(1 - 4x), 0 \leq x \leq 0.5 \\ \sqrt{3}(4x - 3), 0.5 \leq x \leq 1 \end{cases} \tag{4}$$

The Franklin moments of order $(n + m)$ of the image function $f(x, y)$, where $0 \leq x, y \leq 1$ are computed using the Franklin function, which defines these moments.

$$F_{nm} = \int_0^1 \int_0^1 f(x, y)\varphi_n(x)\varphi_m(y)dxdy \tag{5}$$

According to Eq (5), the Franklin function is a type of primary polynomial orthogonal moment. Unlike traditional higher-order polynomial moments, Franklin moments achieve a non-redundant decomposition of the image while maintaining orthogonality. They effectively describe image features using fewer parameters and require only a single polynomial operation. This reduction in computational complexity enhances processing speed.

**Sub-pixel edge detection based on Franklin moments.** Before performing sub-pixel edge detection, it is essential to calculate and compare differences between pixel points in both row and column directions. The resulting sub-pixel edge detections are then compiled. In this paper, we derive Franklin moments based on this approach. The ideal straight-line edge is denoted by L, as illustrated in Fig 3.

The difference between the gray values on both sides of the line L is k, and l is the distance from the origin to the edge. The first three order moments of the one-dimensional step edge

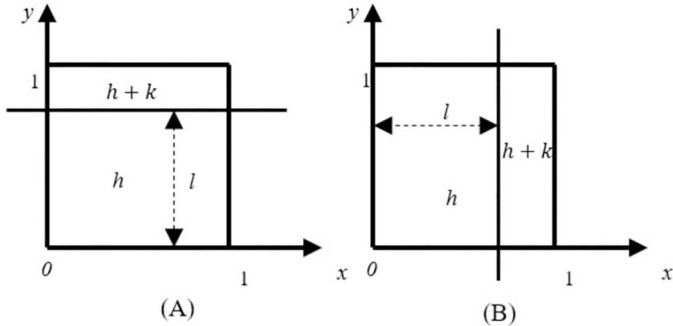

**Fig 3. One-dimensional ideal edge model.** (A) perpendicular to the y-axis, (B)perpendicular to the x-axis. Where (A) and (B) represent the ideal edge conditions of the X-axis and Y-axis, respectively.

can be obtained using Fig 3(a) and Eq (5):

$$F_{00}^1 = h + k(1 - l) \tag{6}$$

$$F_{01}^1 = \sqrt{3}kl - \sqrt{3}kl^2 \tag{7}$$

$$F_{02}^1 = \begin{cases} -\sqrt{3}k(l - 2l^2) & 0 < l \le \dfrac{1}{2} \\ \sqrt{3}k(-1 + 3l - 2l^2) & l > \dfrac{1}{2} \end{cases} \tag{8}$$

When $0 < l < 1/2$, the above equations demonstrate an association:

$$l = \frac{F_{01}^1 + F_{02}^1}{2F_{01}^1 + F_{02}^1} \tag{9}$$

$$k = \frac{\left(2F_{01}^1 + F_{02}^1\right)^2}{\sqrt{3}(F_{01}^1 + F_{02}^1)} \tag{10}$$

$$h = F_{00}^1 - \frac{\left(2F_{01}^1 + F_{02}^1\right)^2}{\sqrt{3}(F_{01}^1 + F_{02}^1)} + \frac{2F_{01}^1 + F_{02}^1}{\sqrt{3}} \tag{11}$$

When $1/2 < l < 1$:

$$l = \frac{F_{01}^1}{2F_{01}^1 - F_{02}^1} \tag{12}$$

$$k = \frac{\left(2F_{01}^1 - F_{02}^1\right)^2}{\sqrt{3}\left(F_{01}^1 - F_{02}^1\right)} \tag{13}$$

$$h = F_{00}^1 - \frac{\left(2F_{01}^1 - F_{02}^1\right)^2}{\sqrt{3}\left(F_{01}^1 - F_{02}^1\right)} + \frac{F_{01}^1\left(2F_{01}^1 - F_{02}^1\right)}{\sqrt{3}\left(F_{01}^1 - F_{02}^1\right)} \tag{14}$$

The subpixel coordinate equation can be derived from Eqs 9–14:

$$\begin{bmatrix} x_s \\ y_s \end{bmatrix} = \begin{bmatrix} x \\ y \end{bmatrix} + \begin{bmatrix} l \\ 0 \end{bmatrix} \tag{15}$$

According to Eq (15), $x_s$ and $y_s$ represent the subpixel coordinates of the image edges, while $x$ and $y$ are the coordinates of image origin. a Considering the magnification effect of the moment template, which is M × M, the subpixel coordinates of the image are given by:

$$\begin{bmatrix} x_s \\ y_s \end{bmatrix} = \begin{bmatrix} x \\ y \end{bmatrix} + \frac{M}{2}\begin{bmatrix} l \\ 0 \end{bmatrix} \tag{16}$$

The formula for the ideal edge's sub-pixel edge in Fig 3(b) is shown in the following equation:

$$\begin{bmatrix} x_s \\ y_s \end{bmatrix} = \begin{bmatrix} x \\ y \end{bmatrix} + \frac{M}{2}\begin{bmatrix} 0 \\ l \end{bmatrix} \tag{17}$$

Currently, the most commonly used moment template sizes are 5x5 and 7x7. Previous research indicates that 7x7 templates yield better edge detection results. However, in this paper, we focus on the operating mechanism and recommend using a 5x5 template for sub-pixel edge detection after preprocessing the image to remove irrelevant information and emphasize the target edge.

This paper presents the relevant calculations for the 5x5 size template of Franklin's moments. The principle and results are outlined below:

Assuming the image f(x, y) = 1 and denoting its convolution template as $M_{nm}$, we can conclude that:

$$M_{nm} = \int_0^1 \int_0^1 \varphi_n(x)\varphi_m(y)dxdy \tag{18}$$

The formula for calculating the value of each row and column in the $M_{nm}$ template is as follows:

$$M_{nm}(i,j) = \int_{\frac{i-1}{5}}^{\frac{i}{5}} \int_{\frac{j-1}{5}}^{\frac{j}{5}} \varphi_n(x)\varphi_m(y)dxdy \tag{19}$$

In Eq 19, $i$ and $j$ represent the number of rows and columns, respectively, both of which range from 1 to 5.

Eqs 5, 16 and 17 demonstrate that sub-pixel edge detection results are solely dependent on $F_{01}$, $F_{02}$, $F_{10}$, and $F_{20}$. As per Eqs 2 and 18, $M_{01}$ and $M_{10}$, $M_{02}$ and $M_{20}$ are transposed, thus

requiring only the calculation of $M_{01}$ and $M_{02}$, which can be obtained from Eq 19.

$$M_{01} = \begin{bmatrix} -0.0554 & -0.0277 & 0 & 0.0277 & 0.0554 \\ -0.0554 & -0.0277 & 0 & 0.0277 & 0.0554 \\ -0.0554 & -0.0277 & 0 & 0.0277 & 0.0554 \\ -0.0554 & -0.0277 & 0 & 0.0277 & 0.0554 \\ -0.0554 & -0.0277 & 0 & 0.0277 & 0.0554 \end{bmatrix} \qquad (20)$$

$$M_{02} = \begin{bmatrix} 0.0416 & -0.0139 & -0.0831 & -0.0139 & 0.0416 \\ 0.0416 & -0.0139 & -0.0831 & -0.0139 & 0.0416 \\ 0.0416 & -0.0139 & -0.0831 & -0.0139 & 0.0416 \\ 0.0416 & -0.0139 & -0.0831 & -0.0139 & 0.0416 \\ 0.0416 & -0.0139 & -0.0831 & -0.0139 & 0.0416 \end{bmatrix} \qquad (21)$$

**Improvement and validation of Franklin moment sub-pixel edge detection algorithm.**
When the moment method is used to detect the edge of the image, each pixel of the image is convolved with the moment template. The theoretical distance $l$, grey level difference $k$, and background grey level $h$ are then obtained using the corresponding formulas. Finally, the edge is determined based on $l$, $k$ and $h$. The specific process is shown in Fig 4(a). However, since the image contains a significant amount of irrelevant information, applying subpixel edge detection directly to the entire image would greatly reduce detection efficiency. Therefore, after preprocessing the image, the approximate position of the edge is determined, allowing for the

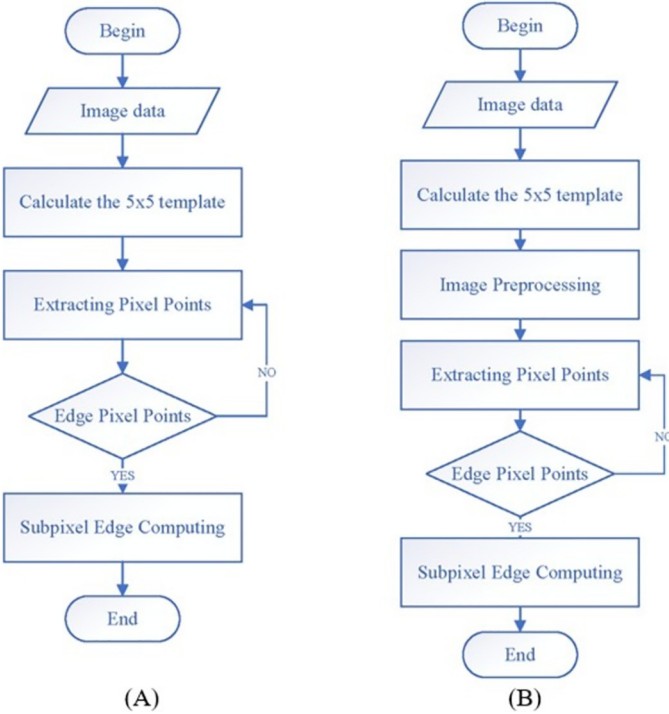

**Fig 4. Edge detection process.** (A) existing process, (B) improved process.

**Table 1. Subpixel method time (unit: ms).**

| Sort | Repulsion Disc | Nozzle |
|---|---|---|
| Franklin Moment | 156 | 4890 |
| Zernike Moment | 322 | 6695 |
| Canny-Zernike | 78 | 2937 |
| Sobel-Zernike | 99 | 3343 |
| Paper Algorithms | 63 | 1841 |

accurate identification of the target contour using the Franklin moment technique. The Laplacian algorithm, which is simple, efficient, and independent of edge direction, is employed to enhance edge sharpness. To improve detection efficiency, the Laplacian algorithm is first used to locate the rough edges, and then the Franklin moment method is applied to refine the subpixel edges. The main steps are shown in Fig 4(b).

The real image of the repulsion disc was used to verify its effectiveness in detecting circuit breaker motion characteristics. The results were compared with those obtained using the Zernike subpixel edge detection algorithm [20], the improved Zernike moment [21, 22], and the initial Franklin moment algorithm from the literature. The outcomes of these different algorithms are shown in Fig 4. To assess the computational efficiency of the proposed algorithm, edge detection was performed on both the local moving image of the repulsion disc (size 172×105 pixels) and the moving image at the nozzle (size 1280×800 pixels). The results of these tests are presented in Table 1.

Fig 5 compares the effectiveness of the algorithm proposed in this paper with other algorithms. The edge detection algorithm based on Franklin moments retains richer image features but is less effective in detecting the motion characteristics of the target. For the research objectives of this paper, the Zernike moments detection method results in messy lines and does not meet detection requirements. This is evident from the results of the Canny-Zernike and Sobel-Zernike algorithms. Although these two methods show improvement over the Zernike moments results, they still suffer from information loss in their edge detection outputs. The proposed algorithm effectively filters redundant information and produces reliable results, surpassing previous algorithms. Table 1 demonstrates a positive correlation between

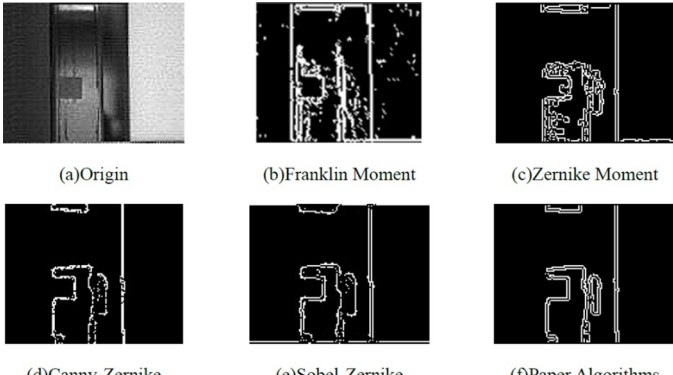

(a)Origin    (b)Franklin Moment    (c)Zernike Moment

(d)Canny-Zernike    (e)Sobel-Zernike    (f)Paper Algorithms

**Fig 5. Comparison of results of different algorithms.** In the images presented: (a) represents the unprocessed initial image; (b) shows the edge image processed using the Franklin moment; (c) displays the edge image processed with the Zernike moment; (d) depicts the edge image processed by the Canny-Zernike method; (e) illustrates the edge image processed by the Sobel-Zernike method; and (f) presents the edge image processed by the algorithm proposed in this paper.

computational efficiency and image size, with larger images resulting in longer computational times. Furthermore, compared to the traditional Zernike algorithm, Canny-Zernike algorithm, and Sobel-Zernike algorithm, the time consumed by the proposed algorithm is reduced by 80%, 30%, 40%, and 60%, respectively. In conclusion, the algorithm proposed in this paper outperforms the traditional sub-pixel algorithms in terms of engineering performance and efficiency.

**Design of a system for characterizing motion.**    In this paper, the motion detection system of the key mechanism of the circuit breaker is constructed based on the above algorithm. By the above algorithm, edge detection is carried out on the moving image of the circuit breaker, the position change of the target point in the image is analysed between adjacent frames, and the position change of the target in real coordinates is obtained by the corresponding proportion conversion, and then the motion characteristics of the circuit breaker are calculated by the kinematic formula. The basic logic flow of motion detection is as follows:

(1) Motion images of the key components of the circuit breaker are collected using the established image acquisition platform.

(2) The collected images are processed using the described method to obtain the edge coordinates of the observed target.

(3) These edge coordinates are then converted into real-world coordinates, and the motion characteristics of the key components of the circuit breaker are calculated using relevant kinematic formulas.

The procedures described in steps (1) and (2) have been detailed above. For step (3), the corresponding calculation methods can be referenced using the speed calculation formula. The real speed of the target is determined by converting pixel coordinates into real-world coordinates, as illustrated in Eq (22).

$$v = \frac{\delta D}{\Delta t}, D = \frac{1}{n} \sum_{i=1}^{n} d_i \tag{22}$$

In Eq (22), $v$ is the velocity in m/s, $\delta$ is the conversion factor between pixel and real size, $\Delta t$ is the time interval between frames, $D$ is the weighted change in pixel points between frames at the target edge in pixels, and $d$ is the change in pixel points between frames at the target edge in pixels.

## Result and discussion

### Image processing result and discussion

The opening motion is crucial for circuit breakers to interrupt faulty circuits and ensure safety. Studying its motion characteristics is vital for evaluating the breaking capacity and timing of circuit breakers. This paper examines the switching motion characteristics of a 252kV circuit breaker under no-load conditions with the control circuit voltage set at 220V. The opening motion involves the movement of the operating structure, followed by the contact movement. The dynamic behavior of the contact nozzle and repulsion disc during the opening motion is illustrated in Figs 6 and 7.

The breaking movement is an important means for circuit breakers to open faulty circuits and ensure the safety of circuits, and its movement characteristics are of great importance in the study of the breaking capacity and breaking time of circuit breakers, so the authors carried out a study of the breaking movement characteristics of 252kV circuit breakers. The tripping movement of the circuit-breaker can be summarised as a process driven by the movement of

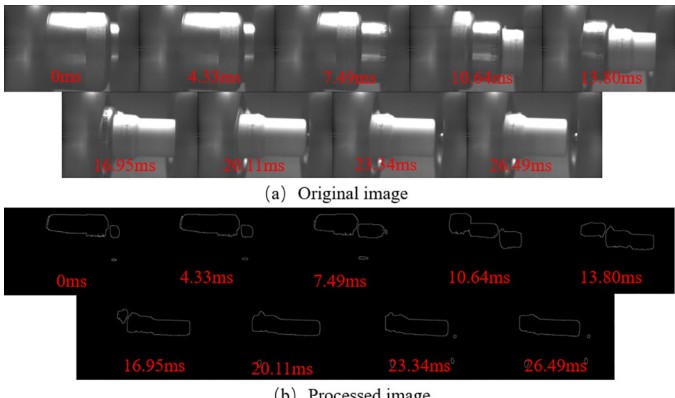

**Fig 6. Excerpt of nozzle movement.** Where (a) is the image of the circuit breaker nozzle movement and (b) is the edge image processed by the algorithm in this paper.

the operating structure, which in turn drives the movement of the contacts; Figs 6 and 7 show the dynamic behaviour of the contact nozzles and the repulsion disc of the operating mechanism during the tripping movement.

As shown in Figs 6 and 7, the algorithm effectively detects the edges of the repulsion disc s. However, it struggles with detecting the overall contour of the nozzle due to the light intensity distribution, although it can detect the edges of the high-intensity areas.

This phenomenon arises because of the differing functions of the two components in the circuit breaker. The repulsion disc generates a reverse electromagnetic force in response to a pulse current, which helps separate the dynamic contacts of the arc extinguisher via the output rod, thus enabling rapid circuit breaking. In contrast, the nozzle serves as a jet channel to disperse and cool the arc. The repulsion disc requires only brief force to alter the arc's direction, while the nozzle must be sufficiently long to facilitate effective arc diffusion and cooling. Additionally, the repulsion disc is designed to be compact for easier installation, whereas the nozzle is longer to create an adequate diffusion channel.

## Motion analysis and discussion

By inputting sequential motion images obtained experimentally into the proposed velocity detection system, it is possible to analyze and calculate the motion velocity of both the repulsion disc and the contact during the tripping operation. The results are presented in Fig 8. Due

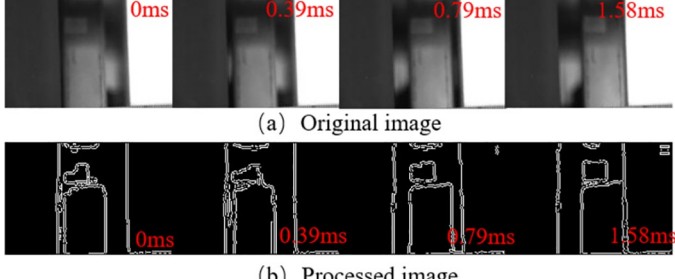

**Fig 7. Excerpt of repulsion disc motion.** Where (a) is the image of the circuit breaker nozzle movement and (b) is the edge image processed by the algorithm in this paper.

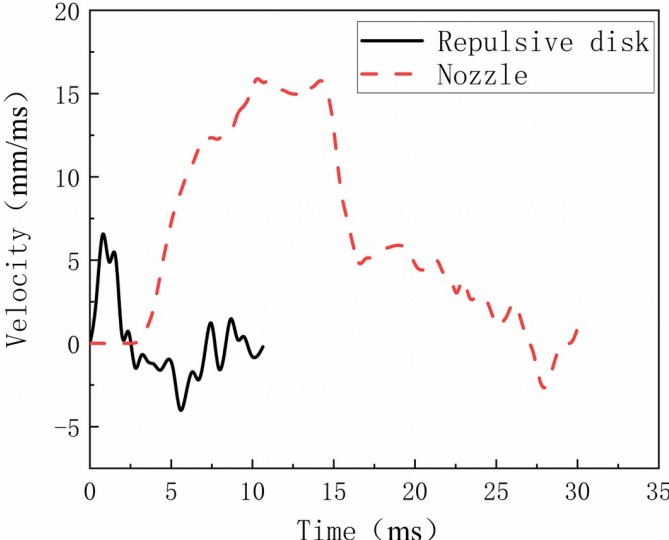

**Fig 8. Velocity of the repulsion plate and the contact opening stage.**

to the prolonged duration of the nozzle bounce, which affects the full display of the repulsion disc 's speed changes, only the speed-time curve of the nozzle from the start to the maximum stroke is shown in Fig 8.

Fig 8 shows that the repulsion disc moves swiftly, starting within 0.23 ms after the control system sends the separation signal, reaching a maximum stroke of 10 mm at 2.36 ms and resetting at 6.38 ms, with a peak velocity of 6.71 mm/ms. The nozzle begins moving within 3.15 ms of the signal, achieving a maximum stroke of 190.05 mm at 26.81 ms, with a peak speed of 16.21 m/s throughout the tripping process. In summary, the repulsion disc completes its movement in 2.36 ms, while the nozzle completes its operation in 26.81 ms. Post-tripping, the contact experiences axial vibration with a maximum amplitude of 5.12 mm, averaging around 2.5 mm, and ceases vibrating by 151.73 ms.

In summary, a noticeable lag is observed between the movement of the repulsion disc and the nozzle. This lag is primarily due to differences in their functions, structures, and materials within the circuit breaker. The repulsion disc acts as the dynamic component, while the nozzle, with its long stroke movement, plays a crucial role in arc guidance and diffusion. Additionally, the lag is affected by the compatibility tolerance between the structural components.

According to Newton's law $F = ma$, the significant force on the repulsion disc during rapid response results from its structural characteristics and operation through electromagnetic induction. The magnetic field produced by the disc, which is proportional to the current in the coil, allows for quick translation of current changes into mechanical motion, enabling rapid arc guidance. The nozzle's movement can be explained by kinetic principles. During its extended stroke, the nozzle is affected by various forces, including the driving force, gas resistance, and structural friction. The interaction of these forces determines the nozzle's trajectory and velocity. Friction affects both the performance and longevity of the nozzle by causing wear and heat. Optimizing friction characteristics can help reduce wear and extend service life.

Additionally, a bouncing phenomenon is observed during the contact opening process, which is a common occurrence in the breaking process of circuit breakers. Its main cause lies in the inertia action and elastic deformation of the nozzle during high-speed separation. When the contacts are rapidly separated under the influence of a strong electromagnetic force, the

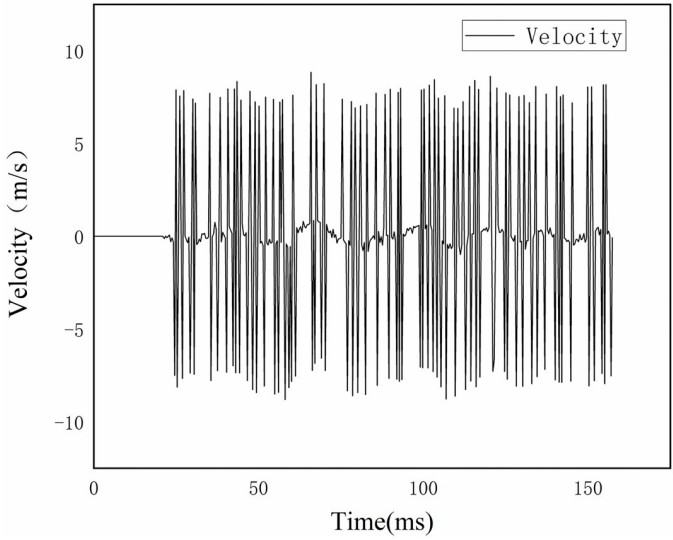

**Fig 9. Velocity of radial vibration of nozzle.**

nozzle is propelled forward momentarily due to inertia and then rebounds due to the presence of elastic elements such as springs. If the rebound height is sufficiently high, there is a possibility of contact re-closure, resulting in the formation of a new arc, increased wear, and diminished performance and lifespan of the circuit breaker. According to data analysis, the maximum bounce stroke is approximately 1/40th of the nozzle stroke, thereby ensuring that the bouncing phenomenon in this type of circuit breaker does not lead to arc reignition, thus allowing for safe circuit interruption.

## Vibration analysis

Research of the experimental data showed that for the nozzle part of the circuit breaker with a large opening distance, in the process of movement by the opening distance and the influence of the driving rod, there will be a certain degree of oscillation phenomenon in its radial direction, and its oscillatory movement is shown in Figs 9 and 10.

As shown in Figs 9 and 10, the nozzle starts to vibrate at 21.37 ms with a vibration interval of [-5.12 mm, 2.05 mm], and the velocity change interval during the whole vibration process is [-8.87 m/s, 8.79 m/s]. This phenomenon signifies that a certain amplitude is reached by the vertical movement of the nozzle. This amplitude is measured in relation to the resting position of the nozzle, with a negative sign denoting movement opposite to the initial position, and a positive sign denoting movement towards the initial position.

In the late opening stage, high-frequency irregular vibrations may be experienced by the nozzle, potentially attributed to aerodynamic effects, mechanical resonance, or thermal expansion and contraction. Additionally, during the transition from high-speed motion to rapid static state, both the moving contact and the driving rod undergo inertial force-induced elastic deformation, storing strain energy in accordance with the law of conservation of energy. This stored energy is subsequently converted into kinetic energy by the elastic restoring force, leading to vibration of the member.

This data is crucial for the study of the dynamic behavior of the nozzle, enabling the identification of the cause and mechanism behind its vibration. If the vibration frequency aligns with the natural frequency of the nozzle, it may be due to mechanical resonance. Conversely, if

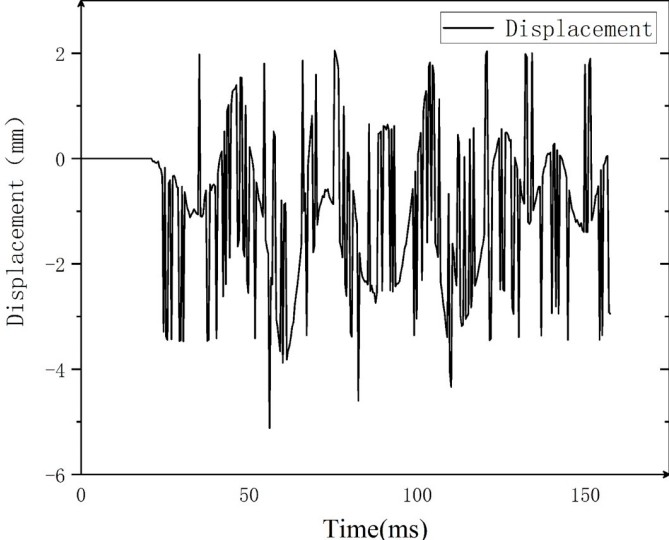

**Fig 10. Vibration displacement of nozzle diameter.**

the vibration is linked to gas flow instability, it may be caused by aerodynamic effects. Furthermore, analyzing the spatial position of the nozzle before and after movement can evaluate its breaking performance in practice. Significant spatial changes in the nozzle's position can impact contact quality upon the next closing and even the service life of the circuit breaker.

Observations from the experiment indicate obvious collision scratches on the nozzle parts, suggesting that vibration significantly affects the nozzle's spatial position, contact quality during closing, and the circuit breaker's service life. Consequently, for high-voltage and high-current circuit breakers, nozzle vibration increases the probability of failure.

## Conclusion

In this work, a circuit breaker motion detection system utilizing Franklin's moments is designed. The study provides a theoretical derivation of the edge detection method based on these moments and verifies the system's superior performance in edge detection. The research combines both experimental and theoretical approaches to detect the tripping motion characteristics of high-voltage circuit breakers, ensuring a comprehensive analysis of their behavior.

Furthermore, this study introduces an edge detection hypothesis based on Franklin moments, deriving an expression that enables the calculation of sub-pixel edge coordinates using a 5x5 convolution kernel. By combining this with the Laplacian algorithm, an edge detection algorithm is developed that shows superior performance. The algorithm's effectiveness and advanced capabilities are validated through rigorous testing.

Finally, based on the foundational formula, this study designs a circuit breaker motion characteristic detection system and establishes a 252kV circuit breaker motion imaging acquisition platform. Experimental validations confirm that the system accurately monitors the motion state of circuit breakers within milliseconds. The findings indicate that high-voltage circuit breakers can react swiftly to tripping signals, completing the disconnection process in just 24 milliseconds. However, significant bouncing and radial vibrations post-tripping are observed, which can adversely affect closure quality and circuit breaker longevity. Overall, the research demonstrates that the proposed method effectively determines fault response times

and motion characteristics, providing essential technical insights for enhancing the understanding of circuit breakers' interrupting capabilities.

## Supporting information

**S1 File. Relevant computed image data.** The relevant data can be obtained from the supporting information. The supporting information is mainly the image data analyzed in this paper. (ZIP)

## Author Contributions

**Conceptualization:** Jinjin Li.

**Data curation:** Zhaoyu Ku.

**Funding acquisition:** Jinjin Li, Huajun Dong.

**Investigation:** Dongheng Li.

**Methodology:** Zhaoyu Ku, Jinjin Li.

**Project administration:** Huajun Dong.

**Software:** Dongheng Li.

**Validation:** Huajun Dong.

**Visualization:** Zhaoyu Ku.

**Writing – original draft:** Zhaoyu Ku.

**Writing – review & editing:** Jinjin Li, Dongheng Li, Huajun Dong.

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
