## [Decision Letter · Decision Letter 0]

25 Jul 2024

PONE-D-24-22027Detection of Opening Motion Characteristics in DC Circuit Breakers Based on Machine VisionPLOS ONE

Dear Dr. Li,

Thank you for submitting your manuscript to PLOS ONE. After careful consideration, we feel that it has merit but does not fully meet PLOS ONE’s publication criteria as it currently stands. Therefore, we invite you to submit a revised version of the manuscript that addresses the points raised during the review process.

**ACADEMIC EDITOR: Please insert comments here and delete this placeholder text when finished.** Be sure to:Authors attempted to detect high-voltage circuit using machine vision with emphasis on opening motion characteristics. First these characteristics and the exact problem of the current methods are missing in the abstract. The result is not clear from the abstract section and authors could attempt to rewrite the abstract to capture the flow of work done exactly Authors may need to expand on the background on the study to provide proper perspective as well as the review of previous similar studies and the gap in a more clear and concise section. The grammatical flow of the entire manuscript requires revision and consistent use of research language. Meanwhile authors presented an improved algorithm flowchart without a proper reference to the existing flow to reflect on the extent of improvement on the new and proposed flow. The methodology section is not really clear to how the machine vision method is used and logical flow into how the results is achieved for motion characteristics opening detection. Overall, it is advised that the entire manuscript should be properly revisited and addressed the aforementioned concerns==============================Please submit your revised manuscript by Sep 08 2024 11:59PM. If you will need more time than this to complete your revisions, please reply to this message or contact the journal office at plosone@plos.org. Please include the following items when submitting your revised manuscript:A rebuttal letter that responds to each point raised by the academic editor and reviewer(s). You should upload this letter as a separate file labeled 'Response to Reviewers'.A marked-up copy of your manuscript that highlights changes made to the original version. You should upload this as a separate file labeled 'Revised Manuscript with Track Changes'.An unmarked version of your revised paper without tracked changes. You should upload this as a separate file labeled 'Manuscript'.

We look forward to receiving your revised manuscript.

Kind regards,

Subhashree Choudhury, Ph.D

Academic Editor

PLOS ONE

Journal Requirements:

2. Thank you for stating the following financial disclosure: "the National Natural Foundation of China (Grant No. 51477023) and the Department of Education of Liaoning Province (Grant No. LJKMZ20220835). "

3. Thank you for stating the following in the Acknowledgments Section of your manuscript: "This study was made possible with the generous support from the National Natural Foundation of China (Grant No. 51477023) and the Department of Education of Liaoning Province (Grant No. LJKMZ20220835). We are grateful for their invaluable financial assistance, which significantly contributed to the successful completion of this research."

Please remove any funding-related text from the manuscript and let us know how you would like to update your Funding Statement. Currently, your Funding Statement reads as follows: "the National Natural Foundation of China (Grant No. 51477023) and the Department of Education of Liaoning Province (Grant No. LJKMZ20220835). "

4. In the online submission form, you indicated that the data underlying the results presented in the study are available from [Pinggao Group Co., LTD]. Data are available from the authors upon request, and with permission from [Pinggao Group Co., LTD].

Additional Editor Comments:

Authors attempted to detect high-voltage circuit using machine vision with emphasis on opening motion characteristics. First these characteristics and the exact problem of the current methods are missing in the abstract. The result is not clear from the abstract section and authors could attempt to rewrite the abstract to capture the flow of work done exactly

Authors may need to expand on the background on the study to provide proper perspective as well as the review of previous similar studies and the gap in a more clear and concise section.

The grammatical flow of the entire manuscript requires revision and consistent use of research language. Meanwhile authors presented an improved algorithm flowchart without a proper reference to the existing flow to reflect on the extent of improvement on the new and proposed flow.

The methodology section is not really clear to how the machine vision method is used and logical flow into how the results is achieved for motion characteristics opening detection.

Overall, it is advised that the entire manuscript should be properly revisited and addressed the aforementioned concerns

Reviewers' comments:

Reviewer's Responses to Questions

**Comments to the Author**

1. Is the manuscript technically sound, and do the data support the conclusions?

Reviewer #1: Partly

Reviewer #2: Yes

2. Has the statistical analysis been performed appropriately and rigorously? 

Reviewer #1: No

Reviewer #2: Yes

3. Have the authors made all data underlying the findings in their manuscript fully available?

Reviewer #1: No

Reviewer #2: Yes

4. Is the manuscript presented in an intelligible fashion and written in standard English?

Reviewer #1: No

Reviewer #2: Yes

5. Review Comments to the Author

Reviewer #1: Authors attempted to detect high-voltage circuit using machine vision with emphasis on opening motion characteristics. First these characteristics and the exact problem of the current methods are missing in the abstract. The result is not clear from the abstract section and authors could attempt to rewrite the abstract to capture the flow of work done exactly

Authors may need to expand on the background on the study to provide proper perspective as well as the review of previous similar studies and the gap in a more clear and concise section.

The grammatical flow of the entire manuscript requires revision and consistent use of research language. Meanwhile authors presented an improved algorithm flowchart without a proper reference to the existing flow to reflect on the extent of improvement on the new and proposed flow.

The methodology section is not really clear to how the machine vision method is used and logical flow into how the results is achieved for motion characteristics opening detection.

Overall, it is advised that the entire manuscript should be properly revisited and addressed the aforementioned concerns

Reviewer #2: Dear author,

1. Why each section looks different font?

2. Proper alignment is missing.

3. What about comparison results with state of art?

4. Improve image quality.

5. Introduction part must be improved.

6. PLOS authors have the option to publish the peer review history of their article (what does this mean?). If published, this will include your full peer review and any attached files.

Reviewer #1: No

Reviewer #2: **Yes: **KARTHIKEYAN B

---

## [Author Response · Author response to Decision Letter 0]

18 Sep 2024

Original Manuscript ID: PONE-D-24-22027

Article Title: “Detection of Opening Motion Characteristics in DC Circuit Breakers Based on Machine Vision”

To: Measurement Editor

Re: Response to reviewers

Dear editor:

On behalf of my co-authors, we thank you very much for giving us an opportunity to revise our manuscript, we appreciate editor and reviewers very much for their positive and constructive comments and suggestions on our manuscript.

We have read reviewer’s comments carefully and have made revision which marked in red in the paper. We have tried our best to revise our manuscript according to the comments. Attached please find the revised version, which we would like to submit for your kind consideration. We would like to express our great appreciation to you and reviewers for comments on our paper. We are uploading our point-by-point response to the comments (below) (response to reviewers).

Looking forward to hearing from you. 

Thank you and best regards.

Prof. Huajun Dong

School of Mechanical Engineering, Dalian Jiaotong University 

E-mail: Xjzbzz@djtu.edu.cn

Reviewer #1: 

Question 1 : Authors attempted to detect high-voltage circuit using machine vision with emphasis on opening motion characteristics. First these characteristics and the exact problem of the current methods are missing in the abstract. The result is not clear from the abstract section and authors could attempt to rewrite the abstract to capture the flow of work done exactly

Author response: The authors are grateful to the reviewer for fruitful advice.The author has revised the abstract to address the issues raised, focusing on clearly presenting the characteristics of circuit breaker motion and highlighting the limitations of current methods

Details：(1)“The detection of high-voltage circuit breaker opening motion has been achieved rapidly and accurately by employing a motion detection method based on Franklin moments.” was revised to “A circuit breaker is a crucial component in power systems, and its operation is essential for evaluating its interruption performance. However, electromagnetic interference often affects sensor accuracy. To address this issue, this paper investigates a non-contact measurement technique for assessing the motion characteristics of circuit breakers. A motion detection method based on Franklin moments is proposed.”in abstract on page 1.

 (2)” This method allowed for the detection of circuit breaker motion at the millisecond level, facilitating the effective capture of vibration parameters and bounce phenomena. High sensitivity was exhibited, and an instantaneous response was provided, supporting further investigation into the fault-cutting ability of circuit breakers..” was revised to “This method can detect the vibration parameters and bouncing phenomenon of circuit breaker motion machine in millisecond level, and the accuracy is 0.01 mm.. These findings offer valuable insights for future research on circuit breaker performance . ” in abstract on page 1.

Quertion 2 : Authors may need to expand on the background on the study to provide proper perspective as well as the review of previous similar studies and the gap in a more clear and concise section.

Author response: The authors appreciate the detailed comments given by the reviewer. In response to the issues raised in the introduction, the authors have revised it to offer a more accurate perspective on the work presented in the paper. The revised introduction includes a review of previous work and highlights its limitations.

Details:(1) “The velocity of the repulsion force and nozzle movement during the breaking period significantly impacts both the breaking time and performance of the DC circuit breaker] Therefore, investigating the characteristics of repulsion force and nozzle movement in a DC circuit breaker will contribute to comprehending its breaking time and performance.” was revised to “The motion characteristics of the circuit breaker are directly reflected in its breaking condition, which is significant when the breaking performance of the circuit breaker is analyzed. As key components, the repulsion disc and nozzle allow the breaking ability of the circuit breaker to be analyzed from various angles, and the motion characteristics to be studied[. Further research into the motion characteristics of the circuit breaker’s repulsion panel and nozzle is deemed essential for the safe operation of the power system.” in the first paragraph of the introduction on page 1.

(2) “Two categories of methods are currently available for detecting movement of key parts of circuit breakers: sensor-based methods and image-processing methods. Acceleration sensors, displacement sensors, grating sensors, and other sensors are installed on the moving structure using the sensor method. However, the circuit breaker contacts are in a high-voltage environment that is closed off, making it impossible for the sensors to detect their motion characteristics directly. ”was revised to “ The sensor method is considered the most effective approach for detecting the motion characteristics of circuit breakers. The moving structure of the circuit breaker is equipped with various sensors, including acceleration, displacement, and grating sensors. However, it is inevitable that the integrity of the circuit breaker structure will be damaged by the sensor, which will have a seriously detrimental effect on its safe operation. Furthermore, as the nozzle of the circuit breaker is situated in a closed high-pressure environment with strong electromagnetic interference, sensors cannot be installed within this internal environment. Therefore, there is an urgent need for a contactless detection method capable of accurately measuring the motion characteristics of the circuit breaker.” in the second paragraph of the introduction on page 1

(3) “Huajun Dong et al. collected image data with a CCD camera and used image processing technology to study the circuit breaker arc morphology and opening and closing speeds, and obtained the speed and arc characteristics of small-gap vacuum circuit breakers during breaking. The static and dynamic characteristics of the circuit breaker when it was short-circuited were calculated by Kacor, P, and the results were compared with the measurements obtained from the high-speed camera. Experimental and numerical methods were employed by Ruiguang Ma to study the arc phenomenon of the air DC circuit breaker[. The experimental arc image was obtained and the influence mechanism of the arc cavity width on the arc motion was analyzed by comparing it with the numerical results. Sequence images of the expansion process of an energy storage spring in an operating mechanism were studied by Zou C. An image pyramid matching algorithm for sequence similarity detection based on recognition region estimation (SSD-P-E) was proposed. The algorithm was able to detect the expansion and deformation characteristics of the spring during the operation of the circuit breaker.The scholars mentioned above have conducted extensive research on circuit breakers. However, their research has some limitations. They did not directly detect the contact motion characteristics of high-voltage circuit breakers, and their analysis of the images did not consider the reaction time of the contact openings. Additionally, they did not analyze the hysteresis of the contacts during the breaking motion of the circuit breaker.” was revised to “The author in [13] and [14] employed a high-speed camera to collect and analyze images of key components of the circuit breaker to determine the opening and closing speeds. The authors in [15] proposed an algorithm for detecting circuit breaker motion using image block matching of moving images. A high-speed camera records the motion of an auxiliary mark on the tie rod or shaft, and the proposed algorithm analyzes its motion characteristics. The author in [16] combined image technology with finite element simulation. By integrating simulation results with measurements obtained from a high-speed camera, the static and dynamic characteristics of the short-circuit release force of the circuit breaker were analysed. The author in [17] proposed a non-contact testing method for the deformation characteristics of circuit breaker springs using image matching tracking technology. By predicting the target position of subsequent frames based on the positional correlation of preceding and following frames, the recognition area is dynamically adjusted according to these predictions. This approach helps eliminate interference from similar structures, enhancing both the matching speed and accuracy. Given the non-contact nature of circuit breakers, it is not surprising that considerable time and effort have been devoted to researching this area by scholars, resulting in extensive findings. Currently, the focus of research into non-contact motion detection methods for high-voltage circuit breakers is placed on indirect detection. By analyzing key structures of circuit breakers, the motion characteristics are inferred.” in the third paragraph of the introduction on page 1

Question 3 : The grammatical flow of the entire manuscript requires revision and consistent use of research language. Meanwhile authors presented an improved algorithm flowchart without a proper reference to the existing flow to reflect on the extent of improvement on the new and proposed flow.

Author response: The authors appreciate the valuable comments from the reviewer. Because the authors are not native English speakers and lacks attention to detail, there were many formatting errors in the paper. We have carefully corrected the formatting errors throughout the paper.

Details: (1) Figure change in the line 183-184 of page 6

(a) existing process (b) improved process

Fig. 4 Edge detection process. Where (a) is the existing process and (b) is the improved process

(2) “The Franklin moment method is employed to detect sub-pixel edges within the dynamic imagery of the operational mechanism. Subsequently, convolutional analysis is conducted using the check image and template derived from Section 1.2 to ascertain the presence of edges in each pixel” was revised to “When the moment method is used to detect the edge of the image, each pixel of the image is convolved with the moment template. The theoretical distance l , grey level difference k, and background grey level h are then obtained using the corresponding formulas. Finally, the edge is determined based on l, k and h. The specific process is shown in Fig. 4(a).” in the first paragraph on page 6 

Question 4 : The methodology section is not really clear to how the machine vision method is used and logical flow into how the results is achieved for motion characteristics opening detection.

Author response: The authors are grateful to the reviewer for fruitful advice. For the issue that the method did not initially include details on detecting the motion characteristics of the circuit breaker nozzle, the author has supplemented the paper with an additional logical flow to address this.

Details: (1) Explain how machine vision methods are used. “In this paper, the motion detection system of the key mechanism of the circuit breaker is constructed based on the above algorithm. By the above algorithm, edge detection is carried out on the moving image of the circuit breaker, the position change of the target point in the image is analysed between adjacent frames, and the position change of the target in real coordinates is obtained by the corresponding proportion conversion, and then the motion characteristics of the circuit breaker are calculated by the kinematic formula.” has been added to the first paragraph on page 7.

(2) The logical flow of motion feature opening detection. 

“The basic logic flow of motion detection is as follows:

(1)Motion images of the key components of the circuit breaker are collected using the established image acquisition platform.

(2)The collected images are processed using the described method to obtain the edge coordinates of the observed target.

(3)These edge coordinates are then converted into real-world coordinates, and the motion characteristics of the key components of the circuit breaker are calculated using relevant kinematic formulas.” has been added to lines 216-223 on pages 7 and 8.

(3) “The sub-pixel algorithm can be employed to acquire the precise coordinates of the motion of both the repulsor disc and the nozzle. By utilizing these obtained sub-pixel coordinates in conjunction with the time axis, Eq. (22) enables the calculation of their respective velocities.” was revised to “The procedures described in steps (1) and (2) have been detailed above. For step (3), the corresponding calculation methods can be referenced using the speed calculation formula. The real speed of the target is determined by converting pixel coordinates into real-world coordinates, as illustrated in Equation (22).” in lines 224-226 on page 8.

Reviewer:2

Dear author,

Question 1: Why each section looks different font?

Author response: The authors greatly appreciate the reviewer’s useful suggestions. The authors have reviewed and addressed this issue by standardizing the font type and size throughout the document. All sections and headings now use a consistent font, and the formatting has been unified. We have resubmitted the revised manuscript with these corrections.

Question 2: Proper alignment is missing.

Author response: The authors appreciate the detailed comments given by the reviewer. The authors have carefully reviewed the formatting of the paper and made the necessary corrections based on your feedback. Specifically, the alignment of tables, figures, and text has been adjusted to ensure conformity with the journal's formatting guidelines.

Question 3: What about comparison results with state of art?

Author response: The authors appreciate the detailed comments given by the reviewer. the authors compare the edge detection performance and response time of the proposed method with several commonly used edge detection algorithms. As illustrated in Fig. 5, the proposed algorithm demonstrates superior edge profile quality compared to the commonly used methods and the Franklin method, with no missing edges observed in these methods. Furthermore, as shown in Table 1, the proposed algorithm exhibits a higher computation speed than the other methods. Details are as follows:

Details:(1) “The results were compared with the Zernike subpixel edge detection algorithm [20], the improved Zernike moment [21,22], and the initial Franklin moment algorithm in the literature. The results of different algorithms are shown in Figure 4.” was in the last paragraph on page 6.

Fig. 5 Comparison of results of different algorithms In the images presented: (a) represents the unprocessed initial image; (b) shows the edge image processed using the Franklin moment; (c) displays the edge image processed with the Zernike moment; (d) depicts the edge image processed by the Canny-Zernike method; (e) illustrates the edge image processed by the Sobel-Zernike method; and (f) presents the edge image processed by the algorithm proposed in this paper.

(2)” In order to verify the computational efficiency of the proposed algorithm, the edge detection efficiency of the local moving image of the repulsion disk (size 172×105pixel) and the moving image at the nozzle (size 1280×800pixel) were verified respectively, and the results are shown in Table 1.” was in the last paragraph on page 6.

Table. 1 Subpixel method time （Unit： ms）

Sort Repulsion Disk Nozzle

Franklin Moment 156 4890

Zernike Moment 322 6695

Canny-Zernike 78 2937

Sobel-Zernike 99 3343

Paper Algorithms 63 1841

Question 4: Improve image quality.

Author response: The authors appreciate the detailed comments given by the reviewer. The author apologizes for the quality issues in the images, which were blurred and distorted due to magnification. The images have been transformed and corrected using the "https://pacev2.apexcovantage.com/" website, and the updated pictures have been incorporated into the article.

Details:(1)Fig. 1 has been changed in the lines 84-86 on page 2

(a) Repulsor disc (b) Nozzle

Fig. 1 Experimental target im

---

## [Decision Letter · Decision Letter 1]

4 Oct 2024

Detection of Opening Motion Characteristics in DC Circuit Breakers Based on Machine Vision

PONE-D-24-22027R1

Dear Dr. Dong,

We’re pleased to inform you that your manuscript has been judged scientifically suitable for publication and will be formally accepted for publication once it meets all outstanding technical requirements.

Kind regards,

Subhashree Choudhury, Ph.D

Academic Editor

PLOS ONE

Additional Editor Comments (optional):

R-1

Some part of the response was written in Chineese Language rather than English. This is important to note for correction before publishing. Hence response to question 5. This will not require further review as it is more of editorial.

R-2

Thanks for the improvement. Try to format before submission . Though it is for reviewing purpose, the reviewer find uneasy to read the same

The results and discussion part is not sufficient. Kindly do reconsider .

Reviewers' comments:

Reviewer's Responses to Questions

**Comments to the Author**

1. If the authors have adequately addressed your comments raised in a previous round of review and you feel that this manuscript is now acceptable for publication, you may indicate that here to bypass the “Comments to the Author” section, enter your conflict of interest statement in the “Confidential to Editor” section, and submit your "Accept" recommendation.

Reviewer #1: All comments have been addressed

Reviewer #2: All comments have been addressed

2. Is the manuscript technically sound, and do the data support the conclusions?

Reviewer #1: Partly

Reviewer #2: Yes

3. Has the statistical analysis been performed appropriately and rigorously? 

Reviewer #1: N/A

Reviewer #2: No

4. Have the authors made all data underlying the findings in their manuscript fully available?

Reviewer #1: Yes

Reviewer #2: Yes

5. Is the manuscript presented in an intelligible fashion and written in standard English?

Reviewer #1: No

Reviewer #2: Yes

6. Review Comments to the Author

Reviewer #1: Some part of the response was written in Chineese Language rather than English. This is important to note.

Reviewer #2: Thanks for the improvement. Try to format before submission . Though it is for reviewing purpose, the reviewer find uneasy to read the same

7. PLOS authors have the option to publish the peer review history of their article (what does this mean?). If published, this will include your full peer review and any attached files.

Reviewer #1: No

Reviewer #2: No

---

## [Editor Report · Acceptance letter]

8 Jan 2025

PONE-D-24-22027R1 

PLOS ONE

Dear Dr. Dong, 

I'm pleased to inform you that your manuscript has been deemed suitable for publication in PLOS ONE. Congratulations! Your manuscript is now being handed over to our production team.

Kind regards, 

on behalf of

Dr. Subhashree Choudhury 

Academic Editor

PLOS ONE